# Inhibition of the CYP Enzymatic System Responsible of Heterocyclic Amines Bioactivation by an *Asclepias subulata* Extract

**DOI:** 10.3390/plants12122354

**Published:** 2023-06-17

**Authors:** Samaria Lisdeth Gutiérrez-Pacheco, Etna Aida Peña-Ramos, Rebeca Santes-Palacios, Martin Valenzuela-Melendres, Adrián Hernández-Mendoza, Armando Burgos-Hernández, Ramón Enrique Robles-Zepeda, Jesús Javier Espinosa-Aguirre

**Affiliations:** 1Coordinación de Tecnología de Alimentos de Origen Animal, Centro de Investigación en Alimentación y Desarrollo, A.C. Carretera Gustavo Enrique Astiazarán Rosas No. 46, La Victoria, Hermosillo 83304, Mexico; lisdeth.gutierrez@gmail.com (S.L.G.-P.); aida@ciad.mx (E.A.P.-R.); martin@ciad.mx (M.V.-M.); ahernandez@ciad.mx (A.H.-M.); 2Laboratorio de Toxicología Genética, Instituto Nacional de Pediatría, Insurgentes Sur 3700-C, Insurgentes Cuicuilco, Coyoacán, Ciudad de México 04530, Mexico; becky_santes@yahoo.com.mx; 3Departamento de Investigación y Posgrado en Alimentos, Universidad de Sonora, Boulevard Luis Encinas y Rosales SN Centro, Hermosillo 83000, Mexico; armando.burgos@unison.mx; 4Departamento de Ciencias Químico-Biológicas, Universidad de Sonora, Boulevard Luis Encinas y Rosales SN Centro, Hermosillo 83000, Mexico; robles.zepeda@unison.mx; 5Instituto de Investigaciones Biomédicas, Universidad Nacional Autónoma de México, Tercer Circuito Exterior sin Número, Ciudad Universitaria, Ciudad de México 04510, Mexico

**Keywords:** CYP inhibition, heterocyclic amines, chemoprotection, medicinal plant, antimutagens, xenobiotics

## Abstract

*Asclepias subulata* plant extract has previously demonstrated antiproliferative activity and antimutagenicity against heterocyclic aromatic amines (HAAs) commonly found in cooked meat. The objective of this work was to evaluate the in vitro ability of an ethanolic extract from the medicinal plant *Asclepias subulata* extract (ASE), non-heated and heated (180 °C), to inhibit the activity of CYP1A1 and CYP1A2, which are largely responsible for HAAs bioactivation. Ethoxyresorufin and methoxyresorufin *O*-dealkylation assays were performed in rat liver microsomes exposed to ASE (0.002–960 µg/mL). ASE exerted an inhibitory effect in a dose-dependent manner. The half inhibitory concentration (IC_50_) for unheated ASE was 353.6 µg/mL and 75.9 µg/mL for heated ASE in EROD assay. An IC_40_ value of 288.4 ± 5.8 µg/mL was calculated for non-heated ASE in MROD assay. However, after heat treatment, the IC_50_ value was 232.1 ± 7.4 µg/mL. Molecular docking of corotoxigenin-3-*O*-glucopyranoside, one of the main components of ASE, with CYP1A1/2 structure, was performed. Results show that the interaction of corotoxigenin-3-*O*-glucopyranoside with CYP1A1/2s’ α-helices, which are related with the active site and the heme cofactor, may explain the plant extract’s inhibitory properties. Results showed that ASE inhibits CYP1A enzymatic subfamily and may potentially act as a chemopreventive agent by inhibiting bioactivation of promutagenic dietary HAAs.

## 1. Introduction

Colorectal cancer (CRC) is one of the most frequent types of cancer in modern society, and responsible for 881,000 deaths worldwide in 2018 [1]. Several research data have evidenced the association of red meat consumption with CRC [2]. This relationship has been linked to several promutagenic heterocyclic aromatic amines (HAAs) formed when meat is exposed to high cooking temperatures [2,3,4]. After ingestion, HAAs can be bioactivated by the isoforms CYP1A1 and CYP1A2 of the cytochrome P450 system (phase I). These enzymes perform a *N*-hydroxylation of the HAAs exocyclic amine group, which can be further esterified by the phase II enzymes acetyltransferases (NAT’s) or sulfotransferases (SULT’s) [5]. The metabolic products of these reactions can spontaneously suffer heterolytic cleavage and be converted to electrophilic nitrenium ions, with the potential to form covalent adducts with DNA inducing genetic mutations that may lead to cancer initiation [6].

Several phytochemicals in dietary components and medicinal plants have proven their anticarcinogenic properties [7,8,9]. Some of the mechanisms involved are free radical scavenging, regulation of tumor suppressor and cell proliferation gene expression, induction of apoptosis, as well as modulation of xenobiotic metabolizing enzymes (XMEs) [10]. The last is a novel approach in chemoprevention; extracts from medicinal plants have proved chemopreventive potential through XMEs inhibition, such as *Hyptis verticillata* and *Heliopsis longipes,* which reduced the activity of CYP1A1 and CYP1A2 isoforms responsible for mutagens bioactivation [11,12,13].

An extract from the medicinal plant *Asclepias subulata* (ASE), demonstrated by in vitro tests, has anticarcinogenic properties in several cancerous cell lines, including colorectal cancer cells [14]. *Asclepias subulata* is a native Sonoran desert plant from the family *Apocynaceae* that has been used in traditional medicine to treat skin problems, headaches, heart pain, flu, stomach problems and cancer [15]. Recently, the antimutagenic activity of ASE was also determined against HAAs on the *Salmonella typhimurium* reversion assay (Ames test) [16]. According to the scale defined by Negi et al. [17] for antimutagenic potential, ASE exerted a strong antimutagenic activity, since it was higher than 40% inhibition of mutagenicity of heterocyclic amines. Both bioactivities of ASE were associated with its main cardenolide components: calotropin and corotoxigenin-3-*O*-glucopyranoside [16,18].

Inhibition of CYP1A1 and CYP1A2 enzymes seems to be one of the main antimutagenic mechanisms reported [19]. Therefore, it is important to evaluate if the antimutagenic activity found in ASE [16] can be attributed to this mechanism. Additionally, no reports were found on how heating may affect the inhibition of XMEs by medicinal plant extracts. Furthermore, several studies have focused on the evaluation of the potential of raw plant extracts to inhibit the catalytic activity of several XMEs, without considering if their bioactivity can be lost or promoted after heat treatment.

Usually, dietary components and medicinal plants are subjected to high temperatures during food cooking or infusion and tea preparation. In addition, if they are intended to be used as marinates in a food matrix subjected to high temperatures, such as meat, the inhibition potential may be lost. Therefore, the aim of this study was to evaluate in vitro the effect of an *Asclepias subulata* extract (non-heated and heated) to modulate the activities of cytochrome P450 CYP1A1 and CYP1A2 isoforms responsible for HAAs bioactivation. In silico experiments with the main plant extract components and CYP1A1/2 structures were also performed to support activity inhibition as the mechanism behind ASE antimutagenic effect.

## 2. Materials and Methods

### 2.1. Plant Material

Aerial parts of *Asclepias subulata* were collected in the city of Hermosillo (29°8′43.25″ N, 110°57′10.15″ W), Sonora, México. The plant was authenticated by Professor Jesus Sánchez Escalante in the Herbarium of the University of Sonora (voucher specimen no. 17403). Then, the aerial parts were air-dried at room temperature (30 °C), powdered, and stored in a plastic bag for further extraction.

### 2.2. Ethanolic Extract of Asclepias subulata

The ethanolic extract of *Asclepias subulata* was obtained following the method of Jiménez-Estrada et al. [20]. Plant powder (100 g) was macerated in ethanol (1 L) for 10 days at room temperature. The extract was filtered (Whatman paper No. 1), and the solvent was removed under reduced pressure in a rotary evaporator (IKA, RV 10 digital, USA) at 45 °C. The obtained extract was placed in a Pyrex test tube and heated in an oil bath at 180 °C for 3 min, with the purpose of simulating the heating conditions used during meat cooking. Cardenolide content was previously calculated and reported in non-heated (corotoxigenin-3-O-glucopyranoside: 53.1 µg/mg, calotropin: 27.6 µg/mg) and heated (corotoxigenin-3-O-glucopyranoside: 52.3 µg/mg, calotropin: 7.7 µg/mg) ASE [16]. Then, the *Asclepias subulata* extracts (ASEs) (non-heated and heated) were re-suspended in dimethyl sulfoxide for inhibition assays.

### 2.3. Animals

Five male Wistar rats (200–250 g) were provided by the bioterium of the Biomedical Research Institute of the National Autonomous University of Mexico. Animal handling followed the protocols dictated by the Internal Committee for the Care and Use of Laboratory Animals (CICUAL) (ID: 250). Animals were kept on a 12 h light/dark cycle for 5 days (induction period), fed with rodent food and water ad libitum. During induction, each rat was intraperitoneally (i.p.) injected with phenobarbital (30 mg/kg) for 3 days, with an additional dose of 60 mg/kg on the fourth day. Additionally, β-naphthoflavone was injected (i.p.) at 80 mg/kg on day 3 [21]. On day 5, rats were sacrificed by cervical dislocation and livers surgically removed for microsomes preparation.

### 2.4. Preparation of Rat Liver Microsomes

Microsomal fraction was obtained according to Maron and Ames [22]. Rat livers were homogenized with KCl (0.15 M) and centrifuged at 9000× g for 10 min (4 °C) to obtain the S9 fraction (supernatant). Then, the S9 fraction was centrifuged at 105,000× *g* for 60 min (4 °C) and the supernatant was discarded. The pellet was resuspended in a phosphate buffer (32.5 mM KH2PO4, 67.5 mM K2HPO4, pH 7.4), and centrifuged again with the same conditions. Subsequently, the resulting pellet (microsomes) was resuspended in a buffer with 1 mM dithiothreitol, 1 mM EDTA and 20% glycerol and stored at −80 °C for further analysis. Protein content of the microsomal fraction was evaluated by the Bradford method [23] to adjust the concentration of the protein required for the enzymatic assays.

### 2.5. CYP1A1 and CYP1A2 Activities

The ability of the *Asclepias subulata* extract (non-heated and heated) to inhibit CYP1A1 and CYP1A2 activities were determined by the ethoxyresorufin and methoxyresorufin O-dealkylation assays, respectively [24]. Incubation mixture was prepared in a 96-well microplate by adding buffer (50 mM Tris-HCl, 25 mM MgCl2, pH 7.6), substrate (7-ethoxyresorufin, CYP1A1 isoform; 7-methoxyresorufin, CYP1A2 isoform), microsomal protein (80 µg/well, for 1A1 isoform; 150 µg/well, 1A2 isoform), and ASEs (0.002–880 µg/mL). An incubation mixture with the vehicle DMSO (without ASE) was evaluated as a control (100% of CYP1A1/CYP1A2 activities). The microplate was incubated at 37 °C for 3 min. Then, the reaction was initiated by adding 2.5 mM NADPH. Fluorescence measurements were performed every 20 s for 40 min at an excitation and emission wavelengths of 530 and 585 nm, respectively. CYP1A1/A2 activities were calculated with a standard curve of resorufin (5–500 pmol/mL), and results were expressed as a percentage of activity of control.

### 2.6. Molecular Docking

A molecular docking was performed to verify the interaction between CYP1A1 and CYP1A2 isoforms with ASE’s cardenolides (calotropin, and corotoxigenin-3-O-glucopyranoside). The three-dimensional structures of CYP1A1 (PDB: 4I8V) and CYP1A2 (PBD: 2HI4) were obtained from the RCSB Protein Data Bank [25,26]. The Autodock Vina tool was used to calculate the lowest affinity energy (kcal/mol) of the interaction between the proteins and cardenolides using the software USCF Chimera, alpha version 1.16 (University of California, San Francisco, CA, USA). Once the best adjustment (lower energy) was reached, BIOVIA Discovery Studio Visualizer software (v20.1.0.19295 Dassault Systèmes: San Diego, CA, USA) was used to depict the residues and type of interactions. Previously, a validation process with the co-crystalized substrate α-naphthoflavone was performed. The modelling was validated by comparing the binding site of α-naphthoflavone with the crystallographic structure of each isoform. Once the coordinates of the box were obtained, the same conditions were used to dock cardenolides.

### 2.7. Statistical Analysis

To calculate the half inhibitory concentration (IC50), data were fitted in a 4-parameter logistic (4PL) non-linear regression model. IC50 of each extract were calculated with the equation Y = Bottom + (Top − Bottom)/(1 + (IC50/X)^HillSlope) with a confidence interval of 95%. A one way analysis of variance (ANOVA) and Tukey’s comparison test were performed to evaluate the differences in the IC50 data. An IC40 value was calculated when the 50% inhibition was not reached at the concentrations evaluated. Data are expressed as means ± standard deviation. All calculations were obtained with the results of three independent experiments (GraphPad Prism 9.1, La Jolla, CA, USA).

## 3. Results and Discussion

### 3.1. Inhibition of CYP1A1 by ASE (Non-Heated and Heated)

The ability of *Asclepias subulata* extracts (ASEs) (non-heated and heated) to inhibit the ethoxyresorufin-*O*-deethylation (EROD) activity in rat liver microsomes is shown in Figure 1. ASE affects the capacity of CYP1A1 to form resorufin from its ethylated substrate in a dose-dependent manner. Inhibitions were observed between 10 and 800 µg/mL and 0.2–960 µg/mL for non-heated and heated ASE, respectively. The concentration required to inhibit half of the CYP1A1 isoform (IC_50_) activity was 353.6 µg/mL for non-heated ASE. However, after being subjected to heat treatment (180 °C), the IC_50_ was reduced by 79%, demonstrating an increase in the inhibitory activity of ASE (Table 1). This behavior may be attributed to the formation of heat-induced by-products, which may have a higher inhibition ability, or the degradation of molecules that may interfere with the inhibitory effect of ASE. In a previous study, we reported a significant and important decrease in calotropin content from 27.6 ± 2.4 to 7.7 ± 1.7 µg/mg in ASE after heat treating the extract at the same conditions used in this work [16]. It has been reported that steroids or triterpenes, similarly to cardenolides, are usually stable at high temperatures [27]; nevertheless, 72% of calotropin was degraded after heat treatment. No reports of thermal-degradation products of calotropin were found; however, heat may cause a degradation of calotropin into the main two moieties that conform its structure: the steroid carbon-backbone and lactone [28]. These two components may be responsible for the increase in the inhibitory activity of heated-ASE, although no reports were found regarding the inhibitory ability of these components.

It has been well-established that CYP1A1 and CYP1A2 play a major role in the bioactivation of HAAs; thus, the inhibition of these isoforms may reduce the mutations induced by HAAs in *Salmonella typhimurium* strains used in the Ames test. Hence, these results may partially explain the antimutagenic effect of ASE previously reported against the heterocyclic amines 2-amino-1-methyl-6-phenylimiazo [4,5-b]pyridine (PhIP), 2-amino-3,8-dimethylimidazo [4,5-f]quinoxaline (MeIQx) and 2-amino-3,4-dimethylimidazo [4,5-f]quinoline (MeIQ) [16].

Other authors have also suggested that the antimutagenic effect of some phytochemicals is related to the reduction in EROD activity. For example, Jeng et al. [29] explained that the antimutagenic activity of ethanol bee glue extracts against the heterocyclic amine IQ and benzo[α]pyrene assessed with the Ames test was associated with the inhibition of CYP1A1. Similarly, Alvarez-Gonzalez et al. [30] associated the 85% reduction in EROD activity with the antimutagenicity of grapefruit juice against benzo[α]pyrene. To our knowledge, this is the first report of the ability of the medicinal plant *Asclepias subulata* to inhibit CYP1A1. Nonetheless, *Hyptis verticillata* (Lamiaceae), also a medicinal plant, had lower IC_50_ values than ASEs of 7.6 and 1.9 µg/mL for CYP1A1 and CYP1A2, respectively [11]. Further, Rodeiro et al. [12] reported a lower IC_50_ of 21.1 µg/mL of the medicinal plant *Heliopsis longipes* (Asteraceae) extract to inhibit both CYP1A1/2. Moreover, isolated quassinoids such as quassin and neoquassin, triterpene lactones similar to ASE’s cardenolides, have demonstrated significant inhibition against CYP1A1 isoform [31]. The IC_50_ values reported were 3.57 and 4.65 µg/mL for quassin and neoquassin, respectively.

### 3.2. Inhibition of CYP1A2 by ASE (Non-Heated and Heated)

Figure 2 shows the inhibition of methoxyresorufin *O*-demethylation activity by non-heated and heated ASE. The extracts exerted an inhibitory effect from 200 to 880 µg/mL and 0.02–880 µg/mL for non-heated and heated ASE, respectively. Without heat treatment, ASE was not able to reach the concentration for 50% inhibition; therefore, an IC_40_ value was calculated (Table 1). Similarly to EROD, the ability of non-heated ASE to inhibit CYP1A2 in MROD increased after heat treatment, and the IC_50_ was reached. These results support the importance of evaluating the biological activities of phytochemical extracts not only in their raw form, but also after being subjected to cooking temperatures. Probably, the ability of many raw extracts that have been reported as poor or strong xenobiotic inhibitors may have been under or overestimated since they were not tested after a heat treatment. These findings support more in deep studies of ASE as a possible chemopreventive additive, perhaps as a meat marinated ingredient used before and during the cooking process.

It is important to point out that in a previous cytotoxic assay, heated ASE was evaluated at a concentration as high as 200 μg/mL without exerting toxicity in a non-cancerous cell line [16]. IC_50_ of heated ASE for EROD was below this concentration; however, for MROD inhibition, the concentration needed was slightly above the highest non-toxic concentration tested. Moreover, although there are no extensive data in the literature on the toxicity related to cardenolide molecules which, as has been mentioned, are the main compounds in ASE, calotropin’s LD_50_ is 9800 µg/kg administrated by intraperitoneal route in mice [32] and 103 µg/kg by intravenous via for cats (PubChem CID: 16142) [33]. No LD_50_ values for corotoxigenin-3-*O*-glucopyranoside were found. However, for its aglycon, LD_50_ is 1074 µg/kg by intravenous via for cats (PubChem CID: 12302397) [34]. The concentrations of cardenolides used in this work were below their average toxic lethal values reported.

Kim et al. [35] reported inhibition of CYP1A2 activity by ursolic and oleanolic acids, and triterpenes with structural similarities to cardenolides in ASE (Figure 3). The half inhibition exhibited by these triterpene compounds was 161 and 65.5 µg/mL for ursolic and oleanolic acids, respectively. Shields et al. [31] also reported the modulation of CYP1A2 activity by the quassinoids quassin and neoquassin with IC_50_ values of 22.3 µg/mL and 33.3 µg/mL, respectively.

Similar behavior was observed by Shields et al. [31] with quassinoids and by Delgado-Roche et al. [36] with a hydroethanolic extract of the seagrass *Thalassia testudinum* (Hydrocharitaceae). These authors associated this behavior with the difference between amino acid residues within each active site. They reported that CYP1A2 active site is more restricted than CYP1A1, due to the presence of Thr223, which forms a hydrogen bond with Asp320 between the F and I helices. In CYP1A1, the active site is more accessible since Asp320 is replaced by Asn221; therefore, the hydrogen bond cannot be formed, allowing the interaction with the inhibitor. The results showed that ASE can inhibit the activity of the isoforms CYP1A1/2 responsible for HAA’s bioactivation in the presence of its natural substrate. However, further studies on the evaluation with HAAs are needed to confirm the inhibition capacity of ASE’s.

### 3.3. Molecular Docking

Since calotropin was degraded significantly by heat treatment (27.6 to 7.7 µg/mg) [16] and, in preliminary experiments, this cardenolide was not able to exert an inhibitory effect against CYP1A1/2′ activities, molecular docking was performed only for corotoxigenin-3-*O*-glucopyranoside to explain the interaction of ASE with both CYP isoforms. Additionally, docking of oleanolic and ursolic acids (Figure 3C,D, respectively) were conducted, since these compounds, which have a similar structure to cardenolides, as was mentioned, also exerted an inhibitory effect against CYP1A2 isoform [35]. Establishing if these two acids have a similar interaction with at least CYP1A2 isoform than corotoxigenin-3-*O*-glucopyranoside (the heat-resistant cardenolide) may help to provide evidence that this cardenolide was responsible for the inhibitory effect shown by ASE. As was mentioned in the Materials and Methods section, before performing the molecular modelling with the potential CYP inhibitors (corotoxigenin-3-*O*-glucopyranoside, ursolic acid, oleanolic acid), validation with the co-crystalized ligand (α-naphthoflavone) for both isoforms, were performed. This modeling was conducted to validate the conditions and coordinates of the box where the interaction between the substrate with both enzymes was attained. Further, it was performed to corroborate the interacting amino acid residues within the active site, according to published structures of these isoforms [25,26].

Figure 4 and Figure 5 demonstrate that the orientation and interaction of α-naphthoflavone with the active site within both CYP1A1 and CYP1A2, which occurred accordingly to published data.

Figure 6A shows the modelling of the interaction between corotoxigenin-3-*O*-glucopyranoside and CYP1A1. The most energetically favorable binding was −9.5 kcal/mol, which was reached when the interaction with the L helix occurred. This helix is located behind the active site. The interactions between the cardenolide and CYP1A1 were by hydrogen bond (green) with Arg464, and a conventional C-H bond (blue) with Glu460. This interaction site is very important, since the L helix is part of the binding region of the heme group, which is an important group that provides structure to the active site of this enzyme and interacts with the substrate [37]. Corotoxigenin-3-*O*-glucopyranoside can also interact with the enzyme’s F helix, however, with a lower affinity of −7.9 kcal/mol.

On the other hand, for CYP1A2 (Figure 6B), the binding energy with the cardenolide was lower (−7.8 kcal/mol) than with CYP1A1, and the interaction was made in the F helix of this isoform that conforms the parallel surface to the active site [25]. CYP1A2 interacting residues were Glu228, Asn247, Thr229, Asn234 and Pro235, which mainly interacted through hydrogen bonds with the –OH groups of the sugar moiety of corotoxigenin-3-*O*-glucopyranoside. The F helix has an important role in CYP1A2 since it has been associated with substrate access and egress and the regioselectivity of the active site [25].

Although oleanolic and ursolic acids’ inhibition ability against CYP1A1 has not been reported, the molecular docking of these acids with this isoform was also conducted to evaluate if they showed a similar interaction as the cardenolide. Figure 7 shows that, instead of interacting with CYP1A1′s L helix as corotoxigenin-3-*O*-glucopyranoside had, these acids interacted at the F helix of this isoform, with a lower binding energy of −7.8 kcal/mol and −7.5 kcal/mol, respectively, in comparison to the binding energy between the cardenolide with CYP1A’s L helix of −9.5 kcal/mol. However, the binding energy with the F helix was similar between these compounds. Therefore, if oleanolic and ursolic acids can inhibit CYP1A1, perhaps their ability may differ from the inhibitory effect shown by ASE.

As it can be corroborated in Figure 8, oleanolic and ursolic acids had a similar interaction behavior with CYP1A2 as the cardenolide, since these acids mainly interacted with F helix through hydrogen bonds between their –OH and -COOH groups and the hydrophilic residues of CYP1A2′s F helix. The binding energies of oleanolic and ursolic acids with CYP1A2 were −7.7 kcal/mol and −8.2 kcal/mol, respectively, which were very similar to the binding energy of the cardenolide with this enzyme (−7.8 kcal/mol). These results may provide evidence to corroborate the inhibition of CYP1A2 activity by ASE.

From the results in studies we had made on the cancer protective effects of the extract, we had established that it possesses different biological activities, including inhibition of both cancer cell proliferation (LS-180 colorectal cancer cells) and the mutagenicity of heterocyclic amines formed by meat cooking (PhIP, MeIQ, and MeIQx). The inhibitory effect on CYP activity reported here may partially explain the abovementioned properties, although additional antigenotoxic mechanisms could be involved. Further studies with isolated corotoxigenin-3-*O*-glucopyranoside are needed to corroborate the cardenolide’s potential effect in the ASE’s inhibitory activity and establish the inhibition mechanism.

## 4. Conclusions

We can conclude that ASE has the ability to inhibit the activities of CYP1A1 and CYP1A2 in vitro, with a higher effect on CYP1A1. Heating the extract improved the inhibitory ability of ASE for both isoforms. The higher potential of heated ASE may occur as a result of the degradation of molecules that interfere with the inhibitory activity of non-heated ASE, or by the generation of stronger inhibitory structures within heated ASE.

Therefore, ASE shows potential to be considered as a chemopreventive agent against dietary carcinogens which are produced during a cooking process, such as heterocyclic aromatic amines. The use of animal models to test the antigenotoxic and anticarcinogenic in vivo effects of the extract are needed to confirm the results obtained. Additionally, it will be convenient to address the role of the thermal degradation by-products of calotropin on the increased ability of heated ASE to inhibit CYP1A1 and CYP1A2, and if the concentrations of ASE used in this study can be effective in vivo without exerting a toxic effect.

## Figures and Tables

**Figure 1 plants-12-02354-f001:**
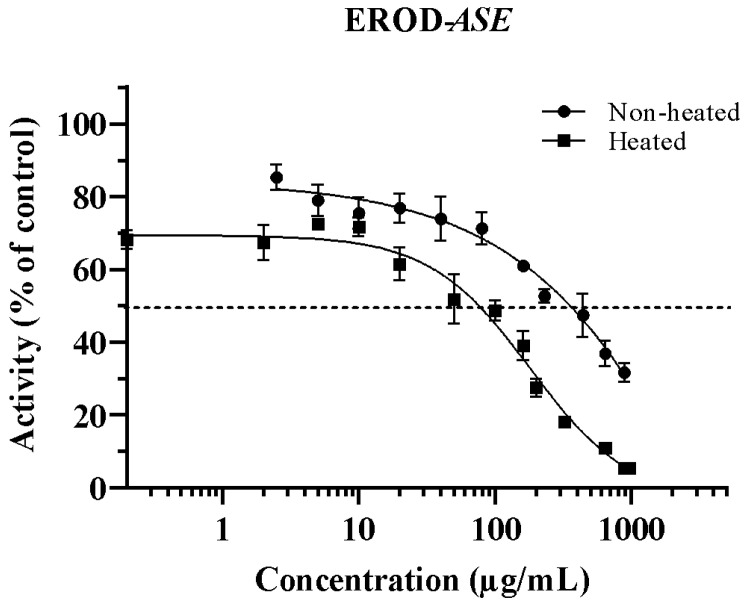
Inhibition of ethoxyresorufin *O*-deethylation activity (EROD) by *Asclepias subulata* extract (non-heated and heated) on rat liver microsomes. Results are expressed as averages ± standard deviation (*n* = 3), *ASE*: *Asclepias subulata* extract. The dotted line represents 50% inhibition. A total of 100% activity of control (incubation mixture without ASE): 217.24 pmol/mg protein/min. ASE was evaluated at 2.5–880 µg/mL (non-heated) and 0.02–960 µg/mL (heated).

**Figure 2 plants-12-02354-f002:**
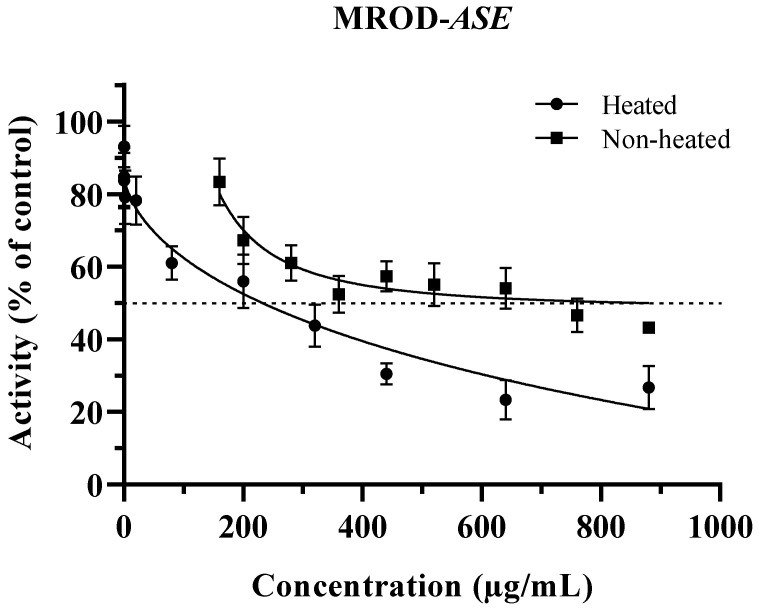
Inhibition of methoxyresorufin *O*-demethylation activity (MROD) by *Asclepias subulata* extracts, non-heated and heated, on rat liver microsomes. Results are expressed as averages ± standard deviation (*n* = 3), *ASE*: *Asclepias subulata* extract. A total of 100% activity of control (incubation mixture without ASE): 9.11 pmol/mg protein/min. ASE was evaluated at 160–880 µg/mL (non-heated) and 0.02–880 µg/mL (heated).

**Figure 3 plants-12-02354-f003:**
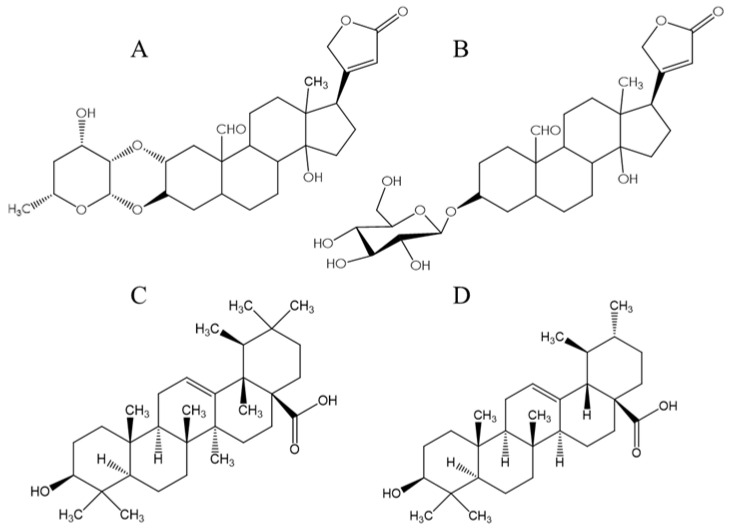
Structures of (**A**) calotropin and (**B**) corotoxigenin-3-*O*-glucopyranoside and similar compounds (**C**) oleanolic and (**D**) ursolic acids.

**Figure 4 plants-12-02354-f004:**
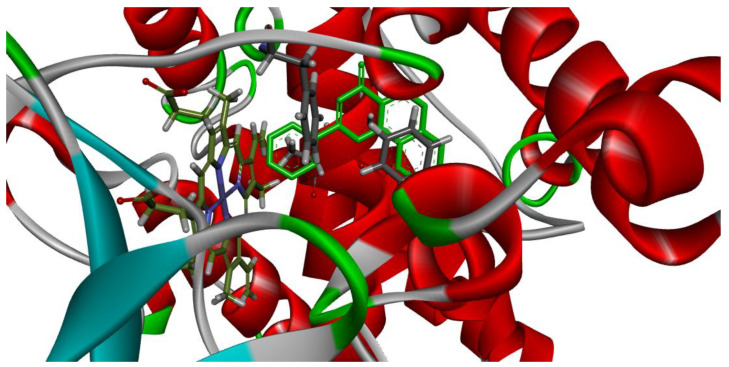
Validation modelling between α-naphthoflavone (light green) and CYP1A1. −14.5 kcal/mol. Center: (−19.6144, 36.1934, −37.9863), size: (44, 46, 49).

**Figure 5 plants-12-02354-f005:**
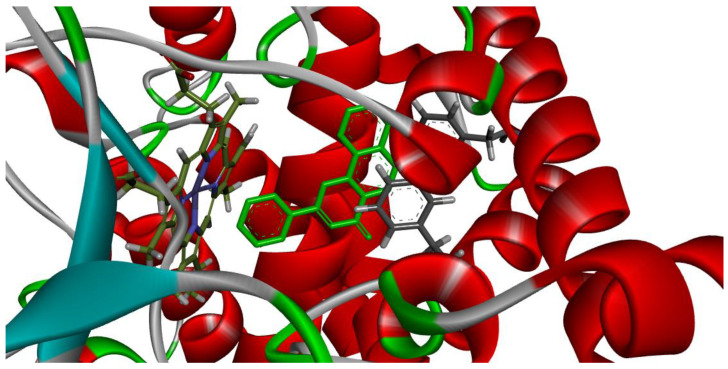
Validation modelling between α-naphthoflavone (light green) and CYP1A2. −14.4 kcal/mol. Center: (3.5573, 20.9096, 30.4746), size: (49.1593, 46.1562, 48.2646).

**Figure 6 plants-12-02354-f006:**
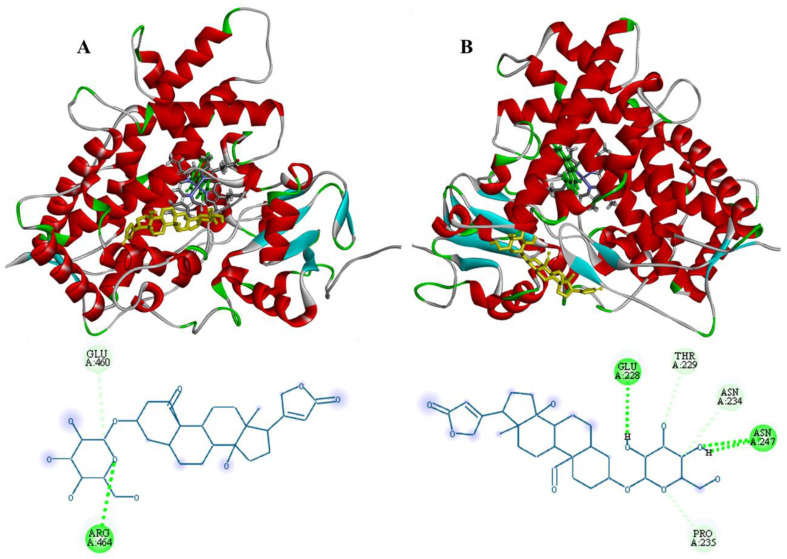
Molecular docking of corotoxigenin-3-*O*-glucopyranoside (yellow) with CYP1A1 (**A**) and CYP1A2 (**B**).

**Figure 7 plants-12-02354-f007:**
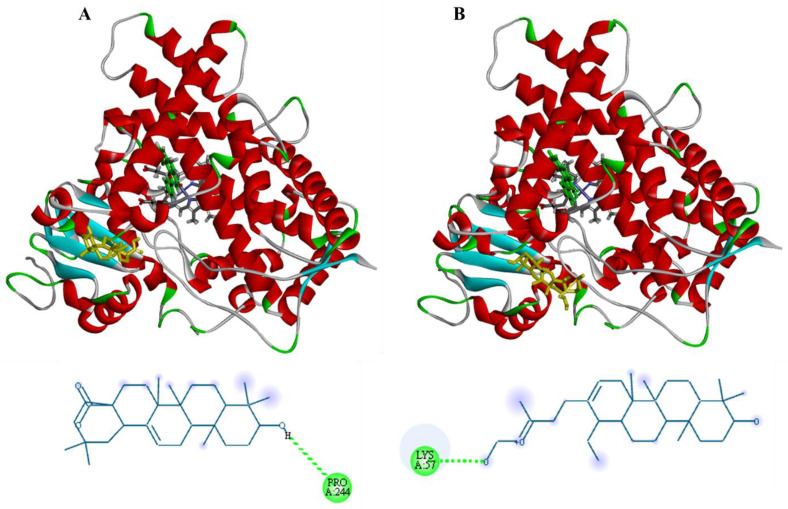
Molecular docking of (**A**) oleanolic and (**B**) ursolic acids (yellow) with CYP1A1.

**Figure 8 plants-12-02354-f008:**
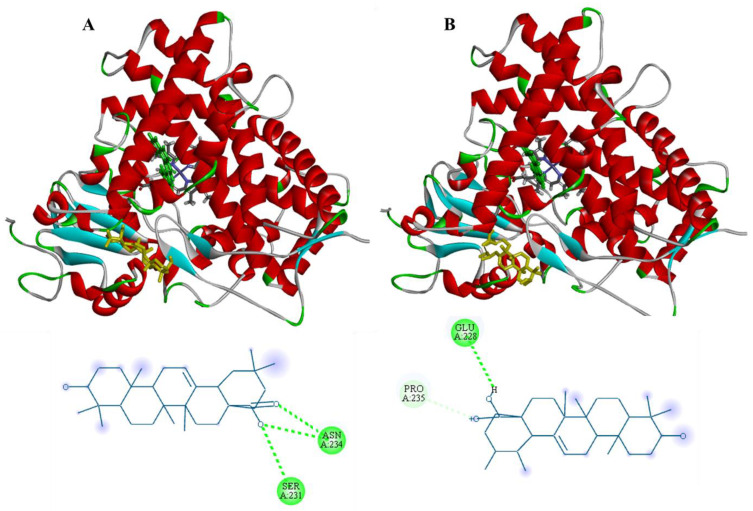
Molecular docking of (**A**) oleanolic and (**B**) ursolic acids with CYP1A2.

**Table 1 plants-12-02354-t001:** IC_50_ values of *Asclepias subulata* extracts (non-heated and heated) on EROD and MROD activity inhibition.

	IC_50_ (µg/mL)	
*Asclepias subulata* Extract	EROD	MROD
Non-heated	353.6 ± 4.51 ^a^	288.4 ^1^ ± 5.81
Heated ^2^	75.92 ± 3.97 ^b^	232.1 ± 7.43

Values are expressed as means ± standard deviation. ^1^ Value expressed as IC_40_. ^2^ Heated at 180 °C for 3 min. EROD: ethoxyresorufin-*O*-deethylation assay. MROD: methoxyresorufin-*O*-demethylation assay. ^a,b^ Different letters in columns represent significant differences *p* < 0.05.

## Data Availability

The data presented in this study are contained within the article.

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
