# Peer review of "Inhibition of the CYP Enzymatic System Responsible of Heterocyclic Amines Bioactivation by an Asclepias subulata Extract"

_plants, 2023, doi:10.3390/plants12122354_

Round 1
Reviewer 1 Report
Dear Authors, Editors,
I would like to ask a few questions and put some comments:
· 25/ ethanolic extract from the medicinal plant Asclepias subulata (ASE)
· 97/ plastic bag and phthalates?
· 107/ (corotoxigenin-3-O-glucopyranoside: 52.3 µg/mg
· 137/ CYP1A1 isoform and CYP1A2 isoform, not 1A1 isoform and 1A2 isoform
- Please, correct in the manuscript (fond type = italics): O-, (corotoxigenin-3-O-glucopyranoside)
- Did you analyze the plant material at the same time, as harvested? What was the ontogenetic stage of the plants in these months?
- Please, see References (title – correct name, grammar)
The Introduction provides a brief description (review) of the pharmacological activity of Asclepias subulate; function of CYP1A1 and CYP1A2 inhibition. The Aims of the manuscript are clearly stated. The Experimental part is concise. The chapter Discussion is appropriate, it compares additional results from several experiments and provides a comparison with previous works.
The submitted manuscript degree contributes to the development of the given topic, meets the requirements put on experimental works of this kind. I recommend this work for publication and assess it positively, after minor revision.
Kind regards
Minor editing of English language required
Reviewer 2 Report
Plants
plants-2432221
Inhibition of the CYP enzymatic system responsible of heterocyclic amines bioactivation by an Asclepias subulata extract
Dear Editor,
The article deals with the evaluation of in vitro ability of Asclepias subulata extract (ASE) on the activity of CYP1A1 and CYP1A2, which are largely responsible of HAAs bioactivation. The topic is good. The manuscript has been well designed and written. My specific comments and questions are below;
- Line 30, 229: IC40 or IC50?
- Line 44: Please give more reference!
- Lines 55 and 61: Please extend this paragraph by giving more information!
- Line 111: Five? Isn’t it very low?
- Line 159: Give statistical design?
- Lines 186 and 188: Give reference!
- The authors are advised to write a conclusion section!
It is good.
Reviewer 3 Report
plants-2432221
Article Inhibition of the CYP enzymatic system responsible of heterocyclic amines bioactivation by an Asclepias subulata extract
Samaria Lisdeth Gutiérrez-Pacheco, E. Aida Peña-Ramos , Rebeca Santes-Palacios , Martin Valenzuela Meléndez , Adrián Hernández-Mendoza , Armando Burgos-Hernández , Ramón Enrique Robles-Zepeda , Javier J. Espinosa-Aguirre *
Gutiérrez-Pacheco et al explored the ability of Asclepias subulate extracts of altering the enzyme activity of cytochrome P450 isoform in rat liver microsomes. They observed that Asclepias subulata extracts can inhibit cytochrome in a dose-dependent manner, demonstrating different behaviors upon heating of the extract.
In this work the potential mechanism of cytochrome P450 inhibition by molecules present in the extract of Asclepias subulata was explored by in vitro and in silico experimental models and a hypothesis based upon results was formulated.
Major and minor comments are reported.
In Materials and Methods, lane 115-119, why were rats injected with phenobarbital? What’s the aim of this treatment? Did authors measure the protein expression of these isoforms of cytochrome P450?
Why did authors use alpha-naphtoflavone as substrate for validation?
Have you ever investigated the effects (cytotoxicity, change in protein expression…) of extracts in cell models?
Did authors evaluate the competitive inhibition between molecules in extract and heterocyclic aromatic amines for the enzyme active site?
Did authors explore the potential molecules in the extracts that can interfere with its inhibitory effect?
In the discussion authors should be point out the need to explore the effects of these extracts in more complex experimental models.
Minor editing of English language required
Round 2
Reviewer 3 Report
Authors addressed all the reviewer’s comments, improving the manuscript as suggested. In my opinion, the paper is suitable for publication.